# CHATSEARCH: A DATASET AND A GENERATIVE RETRIEVAL MODEL FOR GENERAL CONVERSATIONAL IMAGE RETRIEVAL

## ABSTRACT

In this paper, we investigate the task of general conversational image retrieval on open-domain images. The objective is to search for images based on interactive conversations between humans and computers. To advance this task, we curate a dataset called ChatSearch. This dataset includes a multimodal conversational context query for each target image, thereby requiring the retrieval system to infer the underlying retrieval intention from the multimodal dialogue conducted over multiple rounds. Simultaneously, we propose a generative retrieval model named ChatSearcher, which is trained end-to-end to accept and produce interleaved image-text inputs/outputs. ChatSearcher exhibits strong capability in reasoning with multimodal context and can leverage world knowledge to yield more sophisticated retrieval results. It demonstrates superior performance on the ChatSearch dataset and also achieves competitive results on other image retrieval tasks, such as zero-shot text-to-image retrieval and zero-shot composed image retrieval. With the availability of the ChatSearch dataset and the effectiveness of the ChatSearcher model, we anticipate that this work will inspire further research on interactive multimodal retrieval systems.

## 1 INTRODUCTION

Image retrieval is a task that focuses on searching for desired images corresponding to an abstract concept formed in a user's mind, where the user need to somehow convey this concept through human-computer interaction. Various interaction methods have been investigated for image retrieval. One naive interaction approach involves using content related to the desired image, such as a reference image (Tong & Chang, 2001), a set of attributes (Felix et al., 2012) or a descriptive sentence (Li et al., 2011; Radford et al., 2021). Enhanced interaction methods are employed to refine

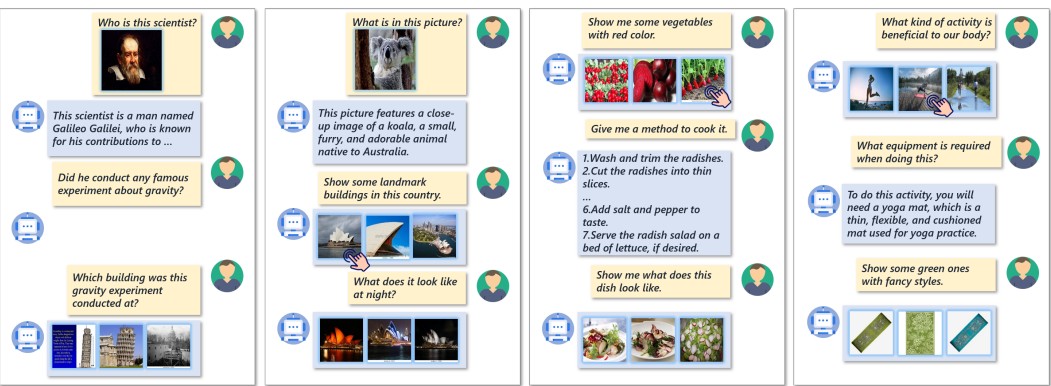

Figure 1: Our generative retrieval model ChatSearcher can accept multimodal inputs and generate textual response or retrieved images. Specifically, user can click for the preferred image results to continue the conversation. ChatSearcher can comprehend multimodal dialogue context, infer user's implicit intentions, generate visual or textual responses through multimodal reasoning and world knowledge, and can also support interactive refinement of results.

image retrieval results by incorporating strategies like relevance score (Rui et al., 1998) or textual user feedback (Liu et al., 2021; Guo et al., 2018; Yuan & Lam, 2021).

The emergence of ChatGPT (OpenAI, 2023a) has shown that a disarmingly simple conversation can serve as an ideal interaction interface for a powerful natural language generation system. Such type of conversation is also an intuitive and effective interaction interface for image retrieval system, offering the following advantages: 1) **Natural interaction**: Users are able to engage with the system using their preferred communication style, resulting in a more natural and seamless retrieval process. 2) **Comprehensive context**: Through conversation, the retrieval system can comprehend and consider the overall context, thereby providing users with more relevant and precise information. 3) **Interactive feedback**: In multi-round conversation, the retrieval results can be progressively refined through interactive user feedback. 4) **Multimodal experience**: Both the user and the system can gather and express information through both image and text, creating a integrated multimodal interaction experience.

In this paper, we investigate the task of general conversational image retrieval on open-domain images. To advance this task, we create a dataset called **ChatSearch**, which necessitates the retrieval model to search for desired images by perceiving a multi-round multimodal dialogue containing both textual and visual human-computer interactions. In ChatSearch, the information needed to retrieve the images is not explicitly stated but often implied within the context of the dialogue. This necessitates the retrieval model to acquire such information through multimodal comprehension, complex reasoning, and world knowledge. To construct ChatSearch, we initially employ a meticulously designed automatic pipeline with the assistance of large-scale models. Subsequently, the dataset undergoes a manual review process conducted by domain experts.

We also introduce a generative retrieval model called **ChatSearcher** specifically designed for conversational image retrieval. ChatSearcher is end-to-end trained to accept interleaved image-text inputs and produce relevant outputs that also combine both images and text in an interleaved format. To accomplish this, we leverage the advanced capabilities of a large language model (LLM). We extract visual embeddings for images in the interleaved input sequence, and concatenate them with textual tokens to form a multimodal token sequence. Specifically, we employ a unified-format training objective for the multimodal sequence, treating both word prediction and image retrieval as generative progresses. In word prediction, we optimize for the probability of ground-truth word prediction within the word vocabulary. For image retrieval, we maximize the probability of image feature matching within a dynamically updated image feature queue, which can be viewed as a visual vocabulary. The training of ChatSearcher involves a two-stage procedure: establishing bidirectional image-text alignment using interleaved image-text data and conversational instruction tuning with diverse instruction data. This instruction data includes conversational image retrieval instructions, visual conversation instructions, and instructions for manipulating AI-generated content (AIGC) images. The derived ChatSearcher model can effectively reasoning out the retrieval query embedding from complex multimodal dialogue context and perform relevance ranking to identify the desired images.

Our paper makes the following contributions:

- We introduce ChatSearch, a dataset for general conversational image retrieval. This dataset emphasizes the need for multimodal reasoning based on multi-round conversations, which is essential for building an intuitive interaction interface for intelligent retrieval systems.

- We propose ChatSearcher, a generative retrieval model that is trained end-to-end to accept and produce interleaved image-text inputs/outputs.

- ChatSearcher demonstrates superior performance on the general conversational image retrieval task. Additionally, it exhibits strong generalization capabilities to other image retrieval tasks, maintaining competitive performances on zero-shot text-to-image retrieval and zero-shot composed image retrieval.

- We will open-source the dataset, the codebase, the model checkpoint, and a conversational retrieval demo to facilitate future research towards interactive multimodal retrieval system.

## 2 RELATED WORKS

### 2.1 IMAGE RETRIEVAL

Image retrieval has been a widely researched topic. Traditional content-based image retrieval (Tong & Chang, 2001) tasks use an image as a query to identify the desired image, with applications spanning product search (Liu et al., 2016), face recognition (Parkhi et al., 2015; Schroff et al., 2015), and image geolocalization (Hays & Efros, 2008). Cross-modal image retrieval introduces external query modalities, such as text-to-image retrieval (Wang et al., 2016; Radford et al., 2021; Jia et al., 2021), sketch-to-image retrieval (Sangkloy et al., 2016), or cross-view image retrieval (Lin et al., 2015), *etc.* Interactive image retrieval (Kushki et al., 2004; Dinakaran et al., 2010; Ferecatu et al., 2008) is raised to refine the image retrieval results with the help of human-computer interaction. Early methods uses simple user feedback to interact with the retrieval system, including relevance (Rui et al., 1998) and attribute (Ak et al., 2018; Han et al., 2017). Some advanced works propose to adopt natural language user feedback to interact (Guo et al., 2018; Liu et al., 2021), which is more familiar with human users. Some works study the image retrieval tasks based on a simple-format human-computer conversation in fashion area (Nie et al., 2021; Yuan & Lam, 2021), plain textual dialogue (Zang et al., 2021) or single utterance in a dialogue (Lee et al., 2022). Some methods try to construct a systematic dialog-retrieval pipeline to enhancing image retrieval (Levy et al., 2023). However, these studies primarily focus on single round interaction or be limited in some special domains like fashion images. In this work, we propose general conversational image retrieval task, aiming to execute image retrieval predicated on an advanced and adaptable form of multimodal conversation in a wider open-domain setting. This necessitates the model to comprehend both general visual and textual context, accommodate the intrinsic user intentions, demanding complex reasoning and the invocation of world knowledge.

### 2.2 MULTIMODAL LARGE LANGUAGE MODELS

With the emergence of ChatGPT (OpenAI, 2023a), human-computer conversational interactions have become a focal point of contemporary discourse. Delving deeper, some researchers have embarked on integrating visual content into dialogues with a Multimodal Large Language Model (Alayrac et al., 2022; Li et al., 2023b; Liu et al., 2023; Zhu et al., 2023a). Recently, studies have explored the intersection between multimodal LLMs and multimodal tasks, including multi-task learning (Dai et al., 2023; Xu et al., 2022; Ye et al., 2023; Gong et al., 2023), multimodal in-context learning (Alayrac et al., 2022; Li et al., 2023a; Monajatipoor et al., 2023), external modality enhancement (Chen et al., 2023; Zhao et al., 2023; Su et al., 2023), dense visual prediction (Wang et al., 2023), multimodal output (Koh et al., 2023b;a; Sun et al., 2023), *etc*.

## 3 CHATSEARCH: A GENERAL CONVERSATIONAL IMAGE RETRIEVAL DATASET

In this section, we explain how we collect the ChatSearch dataset. The common text-to-image retrieval dataset (Lin et al., 2014; Plummer et al., 2015) only require explicit text queries to retrieve the corresponding images. In this paper, we study a more complex situation: retrieving the image based on a multimodal conversation context between human and retrieval system. This task requires the retrieval system to comprehend multimodal contents, as well as extract the retrieval intention from multi-round dialogues. Owing to the absence of existing datasets, we construct a general conversational image retrieval dataset ChatSearch with real-world images.

### 3.1 AUTOMATIC CONSTRUCTION OF GENERAL CONVERSATIONAL RETRIEVAL DATA

The whole automatic data construction pipeline is shown in Fig. 3. Our target is to construct a multimodal dialogue context to search for an image in corpus. During the specific construction process, we primarily expand the existing real-world image-text retrieval dataset MSCOCO (Lin et al., 2014). We utilized existed foundation models including a text generator GPT-4 (OpenAI, 2023b), a gallery retriever CLIP-H (Radford et al., 2021) and a pre-trained image captioner BLIP-2-OPT2.7b (Li et al., 2023b) to assist us in constructing dialogues for image retrieval, instead of relying on external human annotation. After dialogue construction, we apply context merging method to get more complex multimodal dialogues. Finally, we perform manually review on evaluation split data according to the image quality and context relevance in generated dialogues.

**Text dialogue context construction.** We select one raw image caption from the image-text pair in MSCOCO, send it to the text generator GPT-4 and prompt the text generator to generate a multi-round textual dialogue query for image retrieval. We add some constraints in GPT-4 to require that

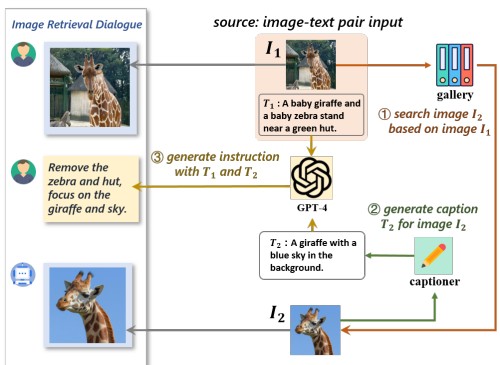 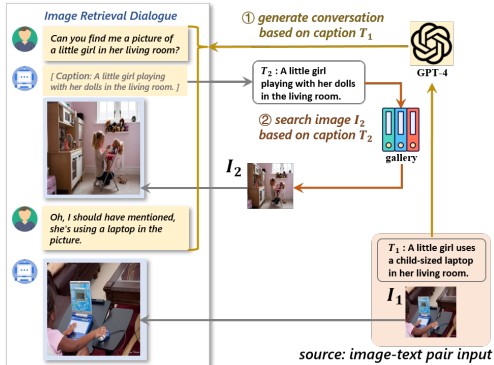

(a) Dialogue construction with reference image. We use the image $I_1$ from source image-text pair as the **reference image** to generate a single-round dialogue.

(b) Dialogue construction with reference text. We use the text $T_1$ from source image-text pair as the **reference text** to generate 2-round multimodal dialogue.

Figure 2: Multimodal dialogue construction. We utilize image-text pairs from the MSCOCO dataset as the source input, select image or caption as a reference point to generate dialogue context. The whole pipeline is combined with a text generator, a image search gallery and a pre-trained image captioner. The final output is a user-assistant multimodal conversation designed for image retrieval. We use numerical indices to represent the execution steps.

the clues for image retrieval are implied within the conversation content. Thus the model needs to reason on the whole dialogue's information and invoke outside-world knowledge to get the correct result. The system prompts for GPT-4 are shown in Appendix.

**Multimodal dialogue context construction.** In this part, we generate multimodal dialogue data for image retrieval, which contains both visual and textual contents. These kind of data require the model to understand the textual user instructions and comprehend the image context as well. We use raw image and image caption as reference image and text to construct dialogues, respectively. The detailed pipelines are shown in Fig. 2 and system prompts used in GPT-4 are shown in Appendix.

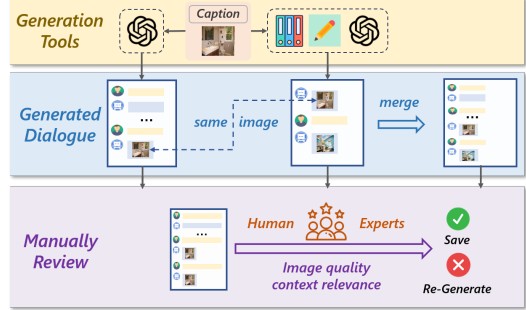

Figure 3: Automatic data construction pipeline. We use foundation models as generation tools to generate textual and multimodal dialogues that aim at searching user desired images, as elaborated in Fig. 2. Then we apply context merging method to get more complex multi-round multimodal dialogue, by merging the plain textual dialogue context and the single-round image-text dialogue which has common images. Finally, we perform manually review on those data to construct a high quality evaluation split.

- **Dialogue construction with reference image.** We generate a single-round multimodal dialogue containing a user-given image, a user-given textual instruction and an assistant-given target image to be retrieved based on user-given image and instruction in Fig. 2a. The dialogue is constructed based on a reference image $I_1$ from the source image-text pair $(I_1, T_1)$ from MSCOCO. We firstly use CLIP-H model to search for the top-10 similar images from the image gallery based on reference image $I_1$ and randomly select one image $I_2$ from the remained images after filtering with a similarity threshold. Then we use the pre-trained image captioner to generate a pesudo target caption $T_2$ for it. Finally, we input the source caption $T_1$ and target caption $T_2$ to the text generator GPT-4 to get the user modification instruction. Constraints are also employed on GPT-4 to highlight the primary difference between two images.

- **Dialogue construction with reference text.** We generate a 2-round image-text dialogue containing two rounds of textual user query and assistant-given target image in Fig. 2b. The dialogue is constructed based on a reference text $T_1$ from the source image-text pair $(I_1, T_1)$ from MSCOCO. We firstly input the reference text $T_1$ to the text generator GPT-4 to acquire a 2-round dialogue while the first round's answer should be a caption $T_2$ representing the visual context in the dialogue and the final answer of the dialogue should be the given target image caption $T_1$.

Then we extract caption $T_2$ from the first round's answer. After that, we search for the most relevant image $I_2$ with caption $T_2$ from the image gallery with CLIP-H model. Finally, we replace extracted caption $T_2$ with retrieved image $I_2$ to construct a 2-round image-text dialogue. Constraints are employed on GPT-4 to remain the semantic connection between two images.

The above reference image and reference text methods bring different natures of the source and target images pair in the constructed dialogue. In dialogues constructed with reference image, the image pair mostly share the same subject, with different background or orientations. In contrary, the image pair in dialogues constructed with reference text have more difference in attributes, varieties and quantities. This result can be attributed to the nature of CLIP embeddings, which primarily emphasize the main object within an image, whereas caption modifications via GPT-4 is more free-form and diverse. Thus, combining these two methods enrich the diversity of the constructed data.

**Dialogue construction with context merging.** After the generation of above data, we merge the plain textual dialogue and the single-round image-text dialogue which has common images together to construct complex interactive dialogues in Fig. 3, providing a more challenging image retrieval task for retrieval systems. To search for the final target image, it need to comprehend the multimodal dialogue context, as well as the user feedback about the previous image result.

### 3.2 BENCHMARK

We select the data sourced from test and val karpathy (Karpathy & Fei-Fei, 2015) split of MSCOCO to compose the ChatSearch test split, while the others are utilized for training. We let five human experts to manually review these test split data and re-generate the unqualified one by following rules. 1) Image quality: Ensuring the retrieved images from LAION-5B are clear, concise, and harmless. 2) Context relevance: Checking whether the multimodal content in the generated dialogue has a reasonable logical and relevance relationship. If experts find the generated data is unsatisfied, they will re-generate the data following above automatic pipeline and intervene with manual adjustments on dialogue contents in each step.

Table 1: Statistics of ChatSearch test split.

| | tChatSearch | iChatSearch | mChatSearch |
|---|---|---|---|
| context modality | text | image-text | image-text |
| sample number | 5000 | 10000 | 10000 |
| average context length | 66.7 | 12.4 | 56.5 |
| dialogue rounds | multi-round | single-round | multi-round |
| task input | textual dialogue between user and assistant | image and text instruction from user | multi-modal dialogue between user and assistant |
| task output | retrieved image candidates | retrieved image candidates | retrieved image candidates |

As shown in Tab. 1, we divided ChatSearch into three sub-tasks: tChatSearch, iChatSearch and mChatSearch, according to the format of dialogue context. tChatSearch is based on the multi-round plain text dialogue context. iChatSearch is based on a single-round image-text context, including reference-image data and reference-text data[1]. mChatSearch is based on a multimodal multi-round dialogue context, which contains complex merged dialogue context and 2-round image-text dialogue generated by reference text.

All sub-tasks are evaluated with recall rate at rank 1, 5, 10. Recall at rank $K$ (R@$K$) quantifies the number of times the correct image is among the top $K$ results. Higher recall means better performance. And we also compute the average recall rate to reflect a general ability on conversational image retrieval.

## 4 CHATSEARCHER: A GENERATIVE RETRIEVAL MODEL

We introduce ChatSearcher, a generative model that is trained end-to-end to accept interleaved image-text inputs and produce relevant outputs that also combine both retrieved images and generated text in an interleaved format.

### 4.1 ARCHITECTURE

As shown in Fig. 4, our model is built with a causal decoder-only LLM, which is initialized with Vicuna-7B v1.5 (Chiang et al., 2023) fine-tuned on an open-source LLM Llama-2 (Touvron et al., 2023). We use OpenAI's CLIP VIT-L (Radford et al., 2021) as the vision backbone to generate visual encodings $f$ and global feature $f_{\langle CLS \rangle}$ for each input image. Then we use a Q-former perceiver to

---

[1]Given the generated 2-round dialogue, we select the image from first-round's answer and the instruction from second-round's question to form a mixed image-text query in iChatSearch task.

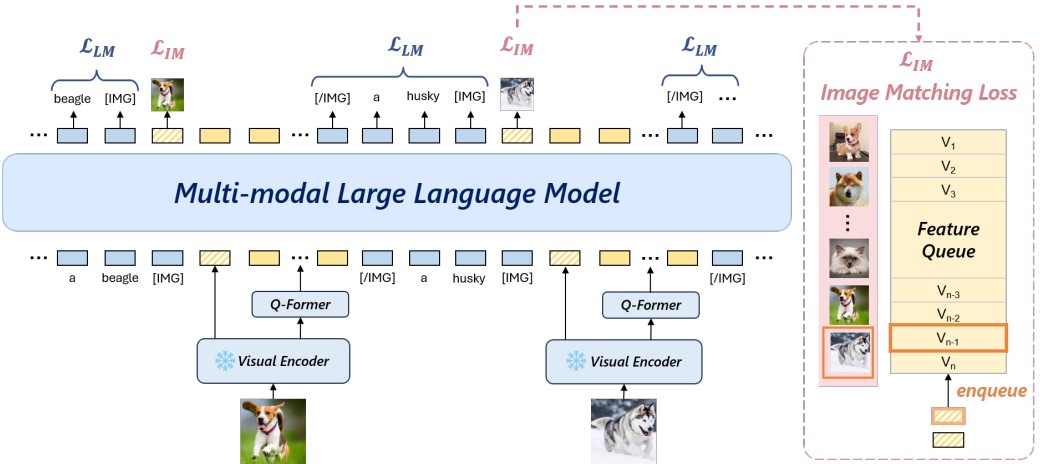

Figure 4: Architecture of our generative retrieval model ChatSearcher. Interleaved documents serve as input, predicting words or retrieving images with generative training objective. Special token [IMG] predicts where to retrieve images. We use a dynamicaly-updated feature queue to save contrastive samples for image retrieval.

compress the visual encodings $f$ from the vision backbone into a fixed number of dense embeddings $q$. The perceiver is initialized with the weight of BLIP-2 Q-former (Li et al., 2023b). And we use different linear projectors to project global visual feature $f_{\langle\text{CLS}\rangle}$ and dense embeddings $q$ into the same dimensions. Then we concatenate them with two special tokens [IMG] and [/IMG] as the visual inputs of the LLM decoder: $[\text{IMG}]\langle\text{CLS}\rangle\langle\text{q}_1\rangle\langle\text{q}_2\rangle...\langle\text{q}_N\rangle[\text{/IMG}]$. We extract the above visual embeddings for each image in the interleaved image-text document and combine them with other textual tokens. We use independent linear head $\mathcal{M}_t$ and $\mathcal{M}_v$ on text output and visual output for text generation and image retrieval respectively.

To enhance the retrieval ability with numerous negative image samples, we build a large sample dictionary that covers a rich set of sample features, which are the global visual feature $f_{\langle\text{CLS}\rangle}$ extracted from the frozen vision encoder. We use a dynamically updated feature queue $\mathcal{Q}$ as the sample dictionary inspired by Wu et al. (2018). The samples in the dictionary are progressively replaced by enqueuing the image features from the current mini-batch and removing the oldest mini-batch.

## 4.2 GENERATIVE TRAINING OBJECTIVE

We use a unified-format training objective for multimodal input sequence, regarding word prediction and image retrieval both as generative progresses. Given an multimodal token sequence $w = \{w_t\}_{t=1}^T$ which contains text tokens and visual tokens, we maximize the likelihood of the ground-truth token:

$$\mathcal{L}(w) = \sum_{w_t \in \mathcal{D}} \log P(w_t|w_{<t}; \theta, \epsilon, \mathcal{Q}) \tag{1}$$

where $\mathcal{D}$ is composed with the multimodal vocabulary except the Q-former feature tokens $\langle\text{q}_{1...N}\rangle$.

The conditional probability $P$ is modeled respectively for image and text tokens in sequence $w$. For text tokens in the sequence, we use the standard language modeling objective, computing the probability that the predicted word is identical to the correct word. For image in the sequence, the model will generate an image retrieval embedding $h_I$ after being queried with the special [IMG] token. Then, we use a linear projector $\mathcal{M}_v$ to project it into the retrieval embedding space $\Phi$. And we use a linear projector $\mathcal{M}_{\mathcal{Q}}$ to project the features in queue $\mathcal{Q}$ into the retrieval embedding space $\Phi$ as well. The normalized cosine similarity for the retrieval query $x$ and sample $y$ in the queue can be computed as:

$$\text{sim}(x, y) = \frac{(x^T\mathcal{M}_v)(y^T\mathcal{M}_{\mathcal{Q}})^T}{\|x^T\mathcal{M}_v\|\|y^T\mathcal{M}_{\mathcal{Q}}\|} \tag{2}$$

And we compute feature matching conditional probability with the queue samples for all image samples in the sequence:

$$P(w_I|w_{<I}; \theta, \epsilon, \mathcal{Q}) = \frac{\exp(\text{sim}(h_I, \mathcal{Q}_{pos})/\tau)}{\sum_{j=1}^{|\mathcal{Q}|} \exp(\text{sim}(h_I, \mathcal{Q}_j)/\tau)} \tag{3}$$

where $\tau$ is the learnable temperature parameter and $I$ represents the index of $\langle\text{CLS}\rangle$ token in the multimodal sequence $w$.

Table 2: General conversational image retrieval results on ChatSearch test split.

| Method | tChatSearch | | | iChatSearch | | | mChatSearch | | | Avg. |
|---|---|---|---|---|---|---|---|---|---|---|
| | R@1 | R@5 | R@10 | R@1 | R@5 | R@10 | R@1 | R@5 | R@10 | |
| random choice | 0.02 | 0.1 | 0.2 | 0.01 | 0.05 | 0.1 | 0.01 | 0.05 | 0.1 | 0.07 |
| CLIP-i (Radford et al., 2021) | - | - | - | 9.65 | 20.96 | 28.05 | 9.65 | 20.96 | 28.05 | 19.55 |
| CLIP-t (Radford et al., 2021) | 15.84 | 34.46 | 45.10 | 14.15 | 30.60 | 39.56 | 12.69 | 27.19 | 35.80 | 28.38 |
| CLIP-ti (Radford et al., 2021) | - | - | - | 12.33 | 26.81 | 35.15 | 13.18 | 28.56 | 37.07 | 25.52 |
| FROMAGe (Koh et al., 2023b) | 15.94 | 36.76 | 48.60 | 12.56 | 29.65 | 39.65 | 14.36 | 32.58 | 42.80 | 30.32 |
| ChatSearcher | **27.38** | **52.48** | **63.50** | **35.54** | **61.16** | **71.57** | **37.90** | **64.22** | **74.06** | **54.20** |

### 4.3 TRAINING METHOD

We use a two-stage training strategy to train our model. More details are shown in Appendix C.2.

**Stage1: Bidirectional Image-Text Alignment.** In the first stage, we try to build the bidirectional image-text alignment (*i.e.* both image-to-text generation and text-to-image retrieval are learned) in our model using interleaved image-text data. We use CC3M (Sharma et al., 2018) and mmc4-core (Zhu et al., 2023b) dataset to pretrain our model. For CC3M image-text pair dataset, we arranged the images and captions in two different configurations: before and after the caption. For mmc4-core interleaved image-text document dataset, we randomly place the image before or after its corresponding sentence and divide the overly long multimodal sequences into several smaller segments at a fixed length for highly efficient training.

**Stage2: Conversational Instruction Tuning.** To help the model better follow the different multi-modal instructions, we construct an instruction tuning dataset with LLaVA-150k (Liu et al., 2023), 10k samples from InstructPix2Pix (Brooks et al., 2023) and 10k samples from ChatSearch training set. LLaVA-150k contains different kind of visual conversation including image-text input and textual response, while InstructPix2Pix and ChatSearch contains images in dialogue's output, which can be formatted as conversational image retrieval instructions furthermore. We employ a question-answer template like "USER :$\langle question \rangle$ ASSISTANT :$\langle answer \rangle$" to unified the instruction format and use a [image] placeholder to represent the image content in the conversation. We compute the text and image loss on answers in each round of the instruction dialogues.

### 4.4 INTERACTIVE INFERENCE

User can interact with ChatSearcher in two ways: providing multimodal instructions or selecting from candidate image results. The model automatically determines whether to output a retrieved image by producing the special [IMG] token based on the multimodal dialogue context. Utilizing the feature embedding of this special query token, the model outputs a set of image candidates ordered by feature similarity. The user can select one of these candidates to continue the interaction. Meanwhile, the selected image is appended to the end of the historical conversation sequence for continuous generation.

## 5 EXPERIMENTS

### 5.1 RESULTS ON GENERAL CONVERSATIONAL IMAGE RETRIEVAL

We evaluate the general conversational image retrieval performance on ChatSearch test split across three distinct tasks: tChatSearch, iChatSearch and mChatSearch. And we compare our model against the baseline model CLIP (Radford et al., 2021) and a Multimodal LLM model FRO-MAGe (Koh et al., 2023b). Given that the CLIP model exclusively accepts either image or text for feature extraction, the performance of CLIP is presented under three configurations: retrieval of images via dialogue text (CLIP-t), retrieval of images through dialogue image features (CLIP-i), and retrieval of images by amalgamating both dialogue text and dialogue image features (CLIP-ti).

As shown in Tab. 4, CLIP shows strong ability in traditional text-based image retrieval. However, it fails in general conversational image retrieval tasks according to the results in Tab. 2. This indicates that while some traditional retrieval models can understand explicit textual expression, they remain limited in comprehending multimodal dialogue content due to a lack of reasoning and knowledge in the perception process. Table 2 shows that ChatSearcher outperforms both CLIP and FROMAGe, suggesting that it possesses superior capability in comprehending image-text interleaved dialogues and discerning the implicit retrieval intentions effectively.

And we also find that for most models, performance on mChatSearch is higher than iChatSearch. Given the overlap in the last-round dialogue between two data parts, it indicates the necessity of incorporating external historical interaction context for better retrieval performance.

Table 3: Zero-shot composed image retrieval (CIR) results on CIRR test set.

| Mode | Method | Recall@K | | | | $R_{subset}$@K | | |
| | | R@1 | R@5 | R@10 | R@50 | R@1 | R@2 | R@3 |
|---|---|---|---|---|---|---|---|---|
| | random choice | 0.04 | 0.22 | 0.44 | 2.18 | 16.67 | 33.33 | 50.00 |
| Fine-tuned | CIRPLANT (Liu et al., 2021) | 19.55 | 52.55 | 68.39 | 92.38 | 39.20 | 63.03 | 79.49 |
| | CompoDiff (T5-XL) (Gu et al., 2023) | 22.35 | 54.36 | 73.41 | 91.77 | 35.84 | 56.11 | 76.60 |
| | CLIP4Cir (Baldrati et al., 2022) | **38.53** | **69.98** | **81.86** | **95.93** | **68.19** | **85.64** | **94.17** |
| Zero-shot | Pic2Word (Saito et al., 2023) | 23.90 | 51.70 | 65.30 | 87.80 | - | - | - |
| | CompoDiff (T5-XL) (Gu et al., 2023) | 19.37 | 53.81 | 72.02 | 90.85 | 28.96 | 49.21 | 67.03 |
| | ChatSearcher | 26.89 | 58.94 | 72.68 | 91.42 | 43.61 | 67.47 | 80.43 |

Table 4: Zero-shot text-to-image retrieval results on Flickr30K and MSCOCO datasets.

| | Flickr30K | | | MSCOCO | | |
| | R@1 | R@5 | R@10 | R@1 | R@5 | R@10 |
|---|---|---|---|---|---|---|
| CLIP (Radford et al., 2021) | **68.7** | **90.6** | **95.2** | 37.8 | 62.4 | 72.2 |
| ChatSearcher (frozen LLM) | 58.5 | 84.4 | 90.3 | 33.7 | 59.6 | 70.5 |
| ChatSearcher (w/o feature queue) | 57.2 | 83.4 | 89.3 | 31.7 | 58.1 | 69.2 |
| ChatSearcher | 68.0 | 87.0 | 92.2 | **41.7** | **67.5** | **76.9** |

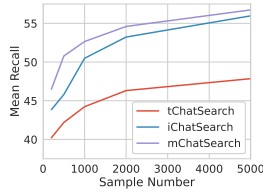

Figure 5: Ablation study on instruction data scale.

## 5.2 Results on Zero-shot Composed Image Retrieval

Composed Image Retrieval (CIR) require the model to retrieve an image according to the reference image and user feedback text. We report the zero-shot retrieval performance of our model on a common used CIR benchmark CIRR (Liu et al., 2021) in Tab. 3. Our model achieves state-of-the-art zero-shot performance on CIRR benchmark. Meanwhile, ChatSearcher also outperform some fine-tuned methods as well. It shows the powerful transfer ability of our model to other image retrieval tasks.

## 5.3 Results on Zero-shot Text-based Image Retrieval

We evaluate the model's zero-shot image retrieval capabilities on two common text-to-image retrieval dataset MSCOCO (Lin et al., 2014) and Flickr30K (Plummer et al., 2015). For evaluation, we use the Karpathy test split of MSCOCO and the test split of FLickr30K, comprising 5k and 1k images, respectively. As shown in Tab. 4, our model's performance is comparable with CLIP, indicating the successful establishment of image-text alignment.

## 5.4 Ablation Study

We perform a comprehensive ablation study on components of ChatSearcher's training process.

**The choice of model design.** We mainly emphasize the importance of feature queue and trainable LLM. A common method to construct negative samples is to collect other image samples within a mini-batch. We compare this method with our feature queue method in Tab. 4. The feature queue provides more negative samples during the computation of feature matching probability, thereby enhancing the model's retrieval capability. We also find that the trainable LLM can effectively improve the retrieval performance in Tab. 4, suggesting that end-to-end training of LLM can improve the model's capacity in comprehending the multimodal dialogue context information.

**Instruction data scale.** In Fig. 5, we explore the impact of scaling instruction data, while the average recall rate on rank 1, 5, 10 for each ChatSearch sub-task is reported. ChatSearcher can achieve better performance compared with CLIP (Radford et al., 2021) with a limited number of instruction data, suggesting a strong multimodal contextual reasoning capability. Moreover, the retrieval performance is improved as the sample number increases, showing a notable scaling trend.

**The importance of visual conversation data.** We add visual conversation data LLaVA-150k (Liu et al., 2023) into our instruction data for two primary objectives: 1) enhancing the conversational

Table 5: Ablation study of the construction of instruction data. *Gray ✓ indicates using only the 5k samples from tChatSearch training set. AIGC represents 10k samples from Instructpix2pix.

| Instruction Data | | | tChatSearch | | | iChatSearch | | | mChatSearch | | | Avg. |
| ChatSearch | LLaVA-150k | AIGC | R@1 | R@5 | R@10 | R@1 | R@5 | R@10 | R@1 | R@5 | R@10 | |
|---|---|---|---|---|---|---|---|---|---|---|---|---|
| ✓ | | | 27.10 | 52.00 | 63.12 | 34.86 | 61.02 | 71.56 | 35.03 | 62.03 | 72.12 | 53.09 |
| ✓ | ✓ | | 27.64 | 52.54 | 63.38 | 35.13 | **61.20** | **71.59** | 35.59 | 62.51 | 72.11 | 53.52 |
| ✓* | ✓ | ✓ | **28.58** | **52.86** | **63.82** | 29.25 | 52.84 | 63.32 | 30.86 | 55.82 | 65.75 | 49.23 |
| ✓ | ✓ | ✓ | 27.38 | 52.48 | 63.50 | **35.54** | 61.16 | 71.57 | **37.90** | **64.22** | **74.06** | **54.20** |

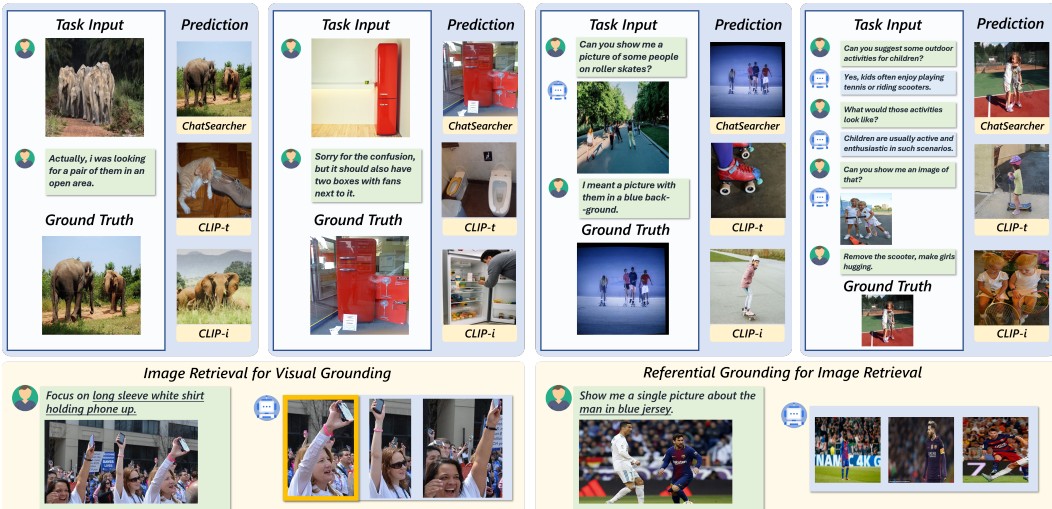

Figure 6: Qualitative results of ChatSearcher. The first row displays ChatSearcher's conversational image retrieval ability across various dialogue contexts, exhibiting superiority over the vanilla CLIP approaches. The second row displays the ChatSearcher's ability in combining grounding and retrieval: using retrieval result to find the region described by a textual phrase, and retrieving images based on a visual reference of the source image.

capability of ChatSearcher for advanced reasoning and 2)bolstering the comprehension capacity of multimodal dialogue context for advanced retrieval. The first requirement is evidenced in conversational samples in Fig. 1 and the second requirement is manifested in the experimental results in Tab. 5. By incorporating LLaVA-150k into our instructional dataset, we attain a superior performance compared to exclusively utilizing ChatSearch data for instruction tuning.

**The effect of adopting AIGC data.** We find that AIGC data for image editing has following characteristics: high similarity between the reference and target images, user modification prompts similar with the feedback in a conversation. This suggests that AIGC data can be structured similarly to conversational image retrieval, which can serve as a valuable augmentation to instructional data. We randomly select 10k synthetic triplet samples from InstructPix2Pix (Brooks et al., 2023) dataset and incorporate them into the instruction dataset . The empirical results in Tab. 5 shows that these AIGC data can also improve model performance and enrich the diversity in our instructional data.

### 5.5 QUALITATIVE RESULTS

As depicted in Fig. 6, we provide a comparison among ChatSearcher and two retrievals method CLIP-t and CLIP-i mentioned in Sec. 5.1. CLIP methods fail to handle the displayed cases, due to their restriction to understanding explicit single-modality expressions. In contrast, ChatSearcher, enhanced by its multimodal reasoning capability, yields precise image retrieval results. Moreover, we find that ChatSearcher has certain image-text grounding capabilities thanks to the alignment built in the first training stage. We explore two way of combining grounding and retrieval: 1) using retrieval result for visual grounding: we use an offline object detector to extract bounding box proposals, subsequently retrieving results from above proposals according to the source image and reference descriptive text. 2)grounded reasoning to retrieval: we rely on the grounding capability of MLLM to retrieve for requests that consists visual reference, such as attributes or orientation, *etc*.

## 6 CONCLUSION

In this paper, we have researched on the general conversational image retrieval task, aiming to extend image retrieval task into a more sophisticated interaction scenario where the retrieval intention is concealed within the multimodal dialogue context. To facilitate this study, we have curated a ChatSearch dataset using a meticulously designed automatic construction pipeline. Additionally, we propose a model called ChatSearcher, which operates under a generative paradigm to retrieve images by reasoning over their multimodal conversational context. ChatSearcher achieves outstanding performance on the general conversational image retrieval task and generalizes well to other zero-shot image retrieval tasks. We anticipate that our work will provide novel perspectives in the fields of image retrieval and human-computer interaction.

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

# A  DISCUSSION

## A.1  UNANSWERABLE SITUATION IN APPLICATIONS.

Assessing user satisfaction with an image is a challenging task, as we are limited in our ability to definitively determine it. Instead, our focus primarily lies in asserting the relative relevance of one image over another. Most previous academic image retrieval tasks (Lin et al., 2014; Plummer et al., 2015) have been proposed to evaluate this ranking ability of the model, asking the retrieval model to rank all candidate images according to their relevance to a given search query. As a academic evaluation dataset, ChatSearch is also proposed to evaluate the ranking ability of the model with a more complex search query hidden in a multi-round multimodal dialogue. Consequently, we still assign ground-truth image for each conversational context to ensure there is no unanswerable situation in evaluation.

However, in practical application scenarios, we still need to deal with this unanswerable situation to enhance user-assistant interaction and improve system robustness. Here we provide two alternative system-level methods.

The first method is to let the model judge if the query is answerable. We set a similarity threshold **p** and subsequently apply this threshold to winnow the retrieved image candidates based on their similarities before presenting them to the user. Only those candidates exceeding the prescribed threshold are retained. If none of image candidates is usable, the model will return a **UNFOUND** response to the user. Additionally, the model will return a predefined response indicating the absence of a suitable answer for the user's query. The model will offer several image candidates with higher similarity to serve as reference points for the user as well.

The second method is to ask the user to judge whether the returned images are usable. To be precise, we add an option labeled as **UNSATISFIED**, which users can choose when none of the returned images prove satisfactory to them during the interaction module.

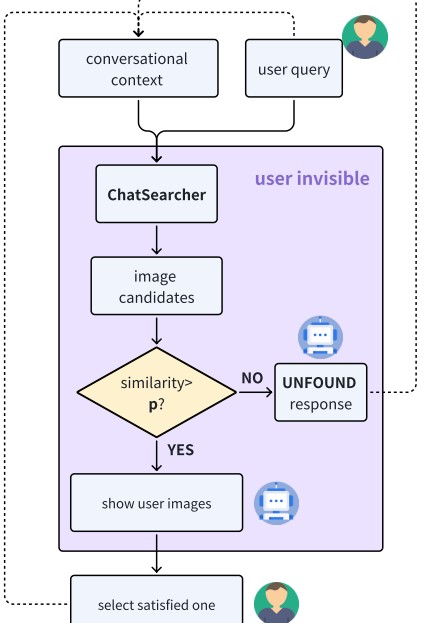 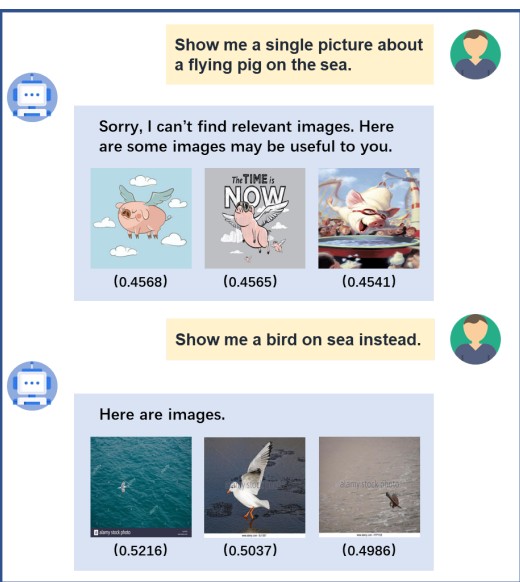

Figure 7: The left sub-figure shows how to let the model judge if the query is answerable. We set a similarity threshold **p** and subsequently apply this threshold to winnow the retrieved image candidates based on their similarities before presenting them to the user. If none of image candidates is usable, the model will return a **UNFOUND** response to the user. In the right sub-figure, we show a example of ChatSearcher with this system method. The numbers indicate the similarity scores computed by ChatSearcher.

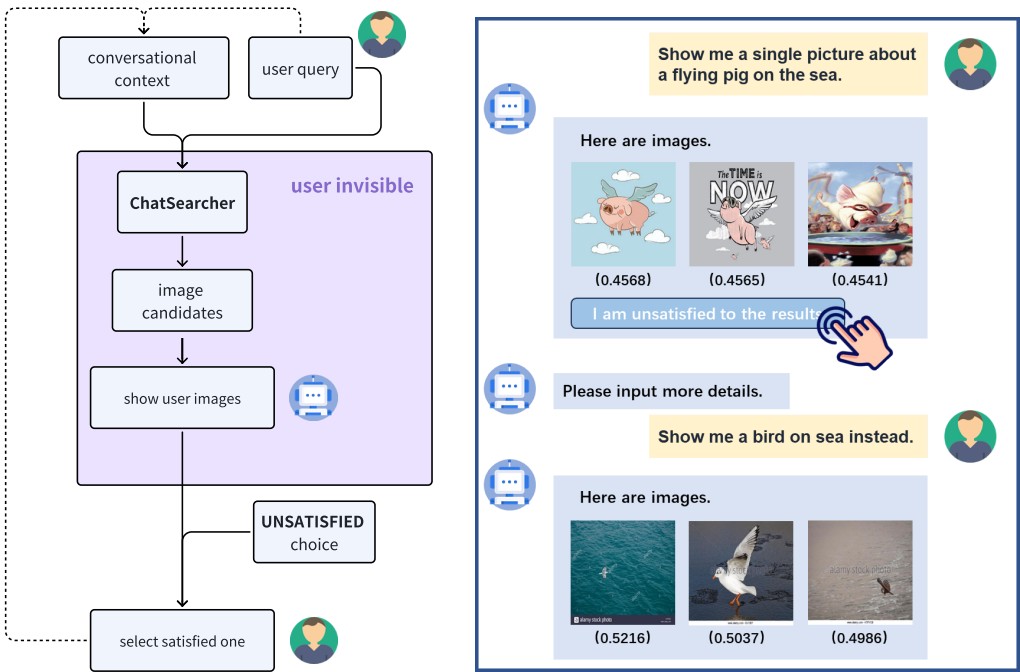

Figure 8: The left sub-figure shows how to ask the user to judge whether the returned images are usable. we add an option labeled as **UNSATISFIED**, which users can choose when none of the returned images prove satisfactory to them during the interaction module. In the right sub-figure, we show a example of ChatSearcher with this system method. The numbers indicate the similarity scores computed by ChatSearcher.

## A.2 ADVANCING CREDIBLE OUTPUTS.

Our dataset and model not only expand the frontiers of interactive image retrieval, but also enhance the essence of multimodal human-computer interactions. Our research offers a novel perspective on enhancing multimodal outputs in human-computer conversations: presenting retrieved factual images to enhance the credibility and clarity of computer-generated information. Looking ahead, we intend to explore the extension of this fact-based credible output approach to diverse modalities including images, videos, audios, *etc*.

## B  PROMPTS FOR GPT-4

In Sec. 3.1, we use GPT-4 as the text generator for data construction. Here we list the system prompts used to constrain the output of GPT-4 in Tab. 6, Tab. 7, and Tab. 8.

Table 6: GPT-4 prompt for text dialogue context construction.

| Input: Target Image Caption |
| --- |
| Prompt:
Assume there's a conversational system that can engage with users and can also retrieve images based on past dialogue content. Given a textual description of a retrieved image, please generate a possible dialogue content. The generated dialogue should follow these criteria:
1.Use "User" and "GPT" to represent the user and the system, respectively.
2.The dialogue should have 3 rounds of Q & A, where the answer in the last round should return the given image.
3.Importantly, the system needs to consider the content of both dialogue rounds to find the specified image, rather than just the last round question.
4.Only the last round is a request to retrieve an image, use pro-noun to represent the content to retrieve.
5.The user's input should always be in the form of questions, and the system's responses should always be statements and no longer than 30 words.
Please generate a dialogue sample following the above points. |

Table 7: GPT-4 prompt for dialogue construction with reference image.

| Input: Source Image Caption and Target Image Caption |
| --- |
| Prompt:
Assuming there's an interactive image retriever that accepts user text input for image searches and can modify previous search results based on user text commands, we need sample data for it. The sample is a tuple of three elements: the description of the returned image from the last round, the user's textual instruction, and the description of the newly retrieved image after considering the user's instruction.
Given descriptions of the original and target images, generate possible user request that:
1.Showing the biggest difference between the original and target images.
2.Emphasize describing this difference rather than prescribing a method to edit.
3.Less than 20 words. |

Table 8: GPT-4 prompt for dialogue construction with reference text.

| Input: Source Image Caption |
| --- |
| Prompt:
Assumpt that there is an AI assistant that helps user to find the desired image by dialog.
You are a dialog designer that generates likely conversations between user and assistant.
Now you are required to generate a 2-turn conversation following the next principle:
1. the conversation should contain 2-turn of user's query and assistant's response.
2. in the first turn, user will give a ambiguous query and assistant should return an image.
3. in the second turn, user will give a supplementary explanation or modification instruction that guide the assistant to find the correct target image and the assistant should return the correct image.
4. when the assistant return the image, it should return a caption that describe the image instead of returning a raw image. Use "[image caption]" to replace the raw image.
5. In the second turn, use pronoun to represent the information from the first turn.
6. In the second turn, only change one object from the first-turn answer.
You are provided with the correct target image which user would like to search. Please generate the dialog. |

# C TRAINING DETAILS

## C.1 DETAILS IN CONVERSATIONAL INSTRUCTION TUNING.

We use instructions from a visual conversation dataset LLaVA-150k (Liu et al., 2023), an image editing dataset Instructpix2pix (Brooks et al., 2023) and a conversational image retrieval dataset ChatSearch for conversational instruction tuning. Since Instructpix2pix is an image editing dataset which contains a triplet of reference image, editing instruction and target image, we simply modify the editing instruction into a more human-like textual instruction with some pre-defined template as shown in Fig. 9. Then we can construct a single-round conversation in which user provide an reference image and an editing instruction to require the model to find the target image with retrieval. We provide samples for each kind of conversational instruction tuning dataset in Fig. 10, which can be all regarded as multi-modal dialogue.

Then, these three kind of datasets are mixed to perform conversational instruction tuning. We employ a question-answer template like "USER :⟨question⟩ ASSISTANT :⟨answer⟩" to convert all conversation data into unified format and use a [image] placeholder to represent the image content in the multi-modal sequence. The multimodal sequence is sent into model for instruction tuning. We compute the text and image loss using generative training objective proposed in Sec. 4.2 on assistant's answers in each round of the instruction dialogues.

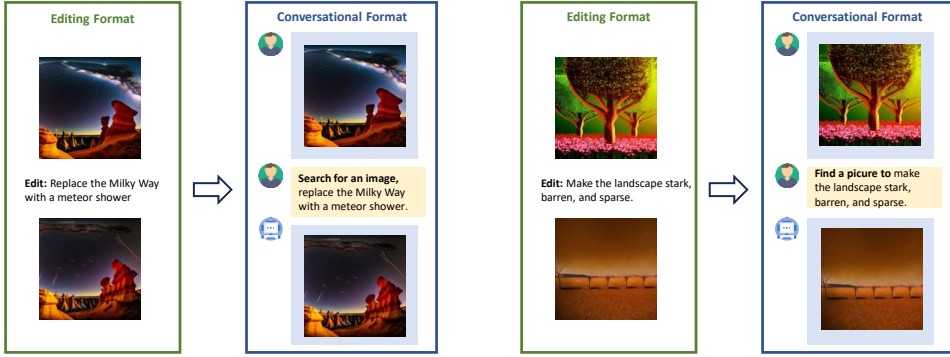

Figure 9: Transform Instructpix2pix sample from editing format to conversational format. We simply modify the editing instruction into a more human-like textual instruction with some pre-defined templates, which are shown in **bold**.

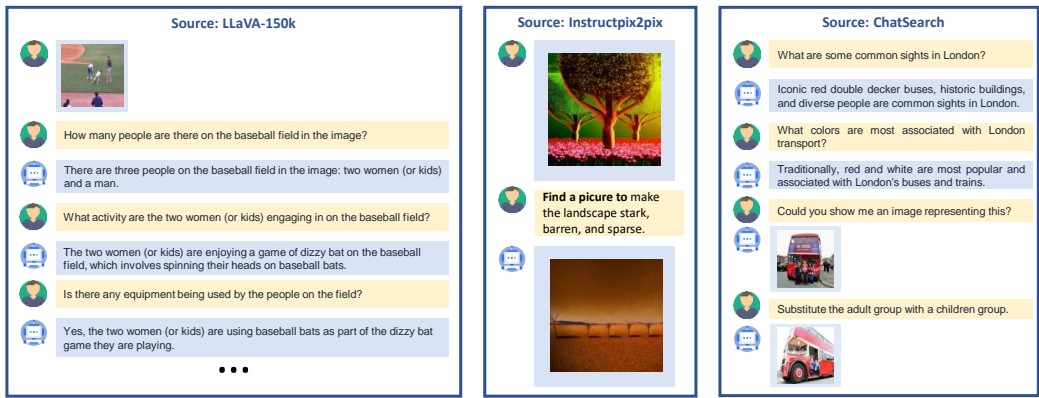

Figure 10: Data sample for visual conversation dataset LLaVA-150k, image editing dataset Instructpix2pix and conversational image retrieval dataset ChatSearch.

## C.2 HYPERPARAMETERS

We list the training details in Tab. 9.

Table 9: Hyperparameters in ChatSearcher Training Procedure

|  | Stage1: Bidirectional Image-Text Alignment | Stage2: Conversational Instruction Tuning. |
|---|---|---|
| training data | CC3M+mmc4-core | ChatSearch-train-10k+LLaVA-150k+Instructpix2pix-10k |
| context length | 350 | 350 |
| batch size | 256(CC3M), 64(mmc4-core) | 64 |
| queue size | 10000 | 1000 |
| image resolution | 224 | 224 |
| | Optimizer Setting | |
| optimizer | AdamW | AdamW |
| $\beta_1$ | 0.9 | 0.9 |
| $\beta_2$ | 0.98 | 0.98 |
| weight decay | 0 | 0 |
| | Scheduler Setting | |
| learning rate scheduler type | cosine | cosine |
| peak learning rate | 2e-5 | 2e-6 |
| warm up ratio | 0.03 | 0.03 |
| training steps | 200k | 10k |

# D CONVERSATIONAL SAMPLES

## D.1 INTERACTION BRANCHES SAMPLES

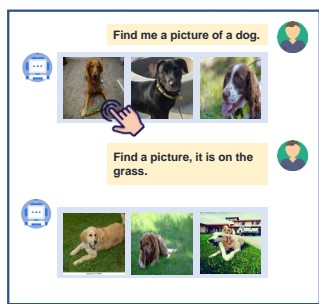 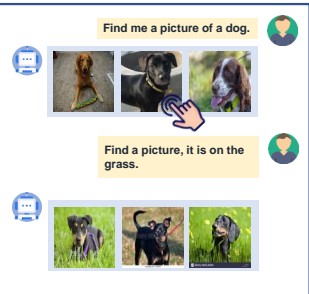 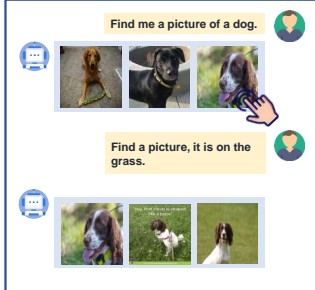

Figure 11: Interaction branches on **result choosing**. We show that **different choices on image results** in previous round can influence the results in following round. In these samples, user choose different image returned by ChatSearcher and input same instruction to interact with model. ChatSearcher return different results based on user's choice and instruction.

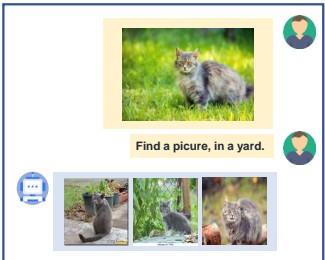 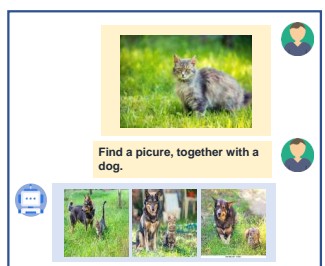 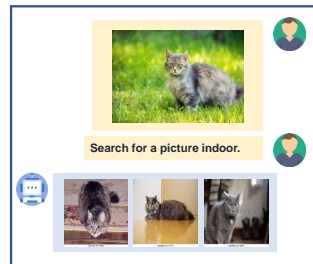

Figure 12: Interaction branches on **instruction choosing**. We show that **different text instructions** on same image can influence the results. In these samples, user input different instructions with a same image. ChatSearcher return different results based on user's instruction and given image.

## D.2 MULTI-ROUND CONVERSATIONAL SAMPLES

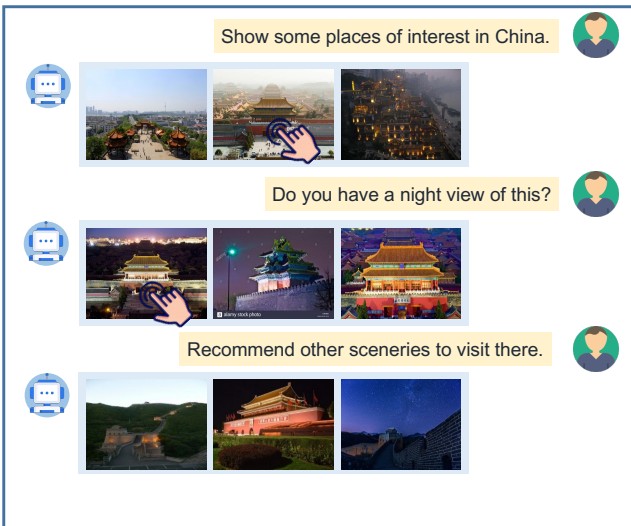

Figure 13: Multi-round conversational sample of ChatSearcher.

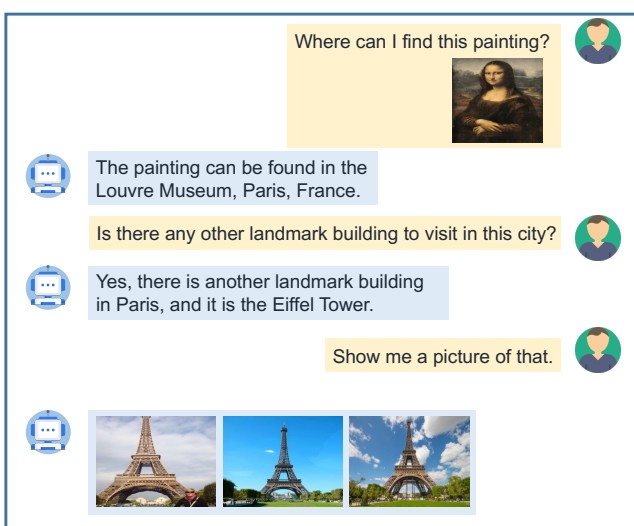

Figure 14: Multi-round conversational sample of ChatSearcher.

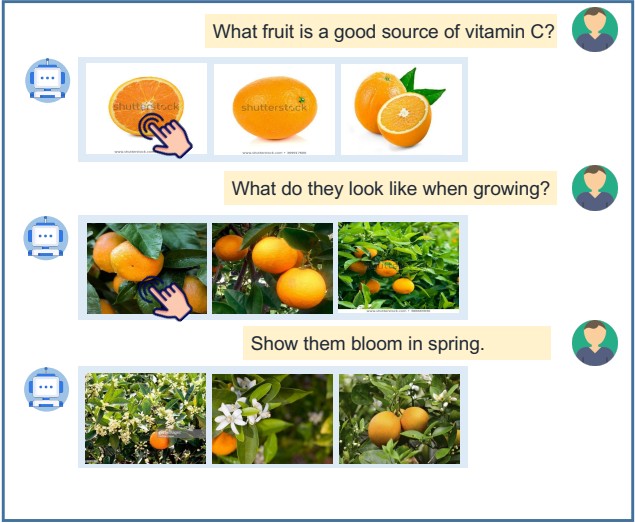

Figure 15: Multi-round conversational sample of ChatSearcher.

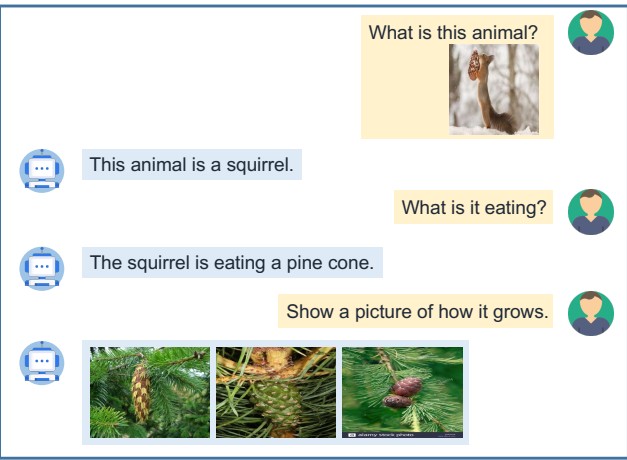

Figure 16: Multi-round conversational sample of ChatSearcher.

# E    MORE QUALITATIVE RESULTS

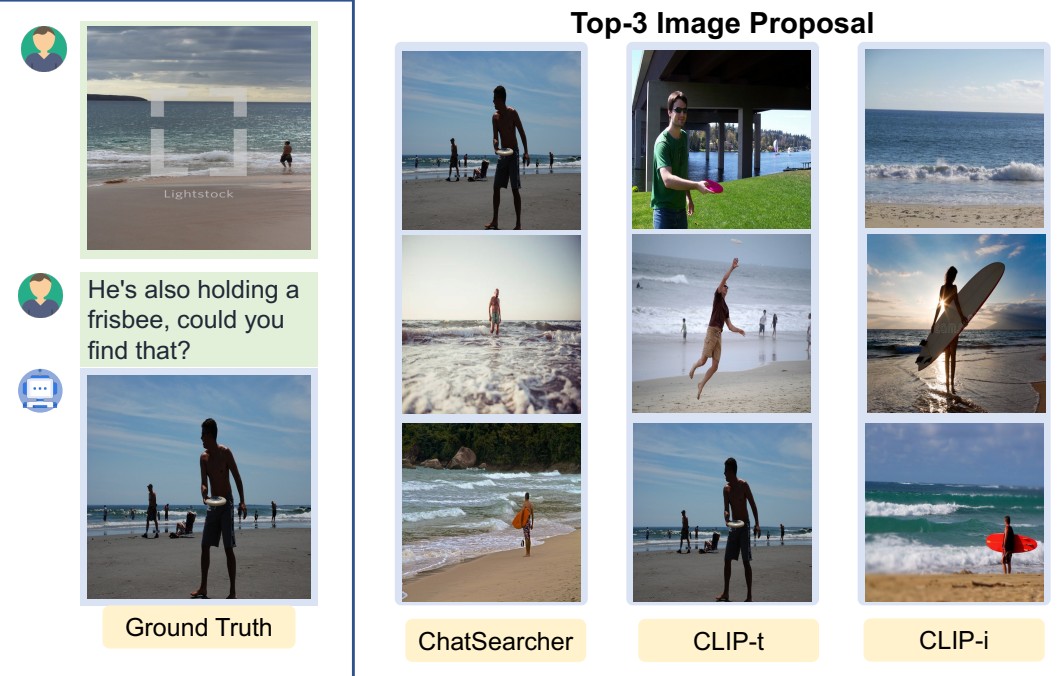

Figure 17: Visualize comparation between ChatSearcher and other methods.

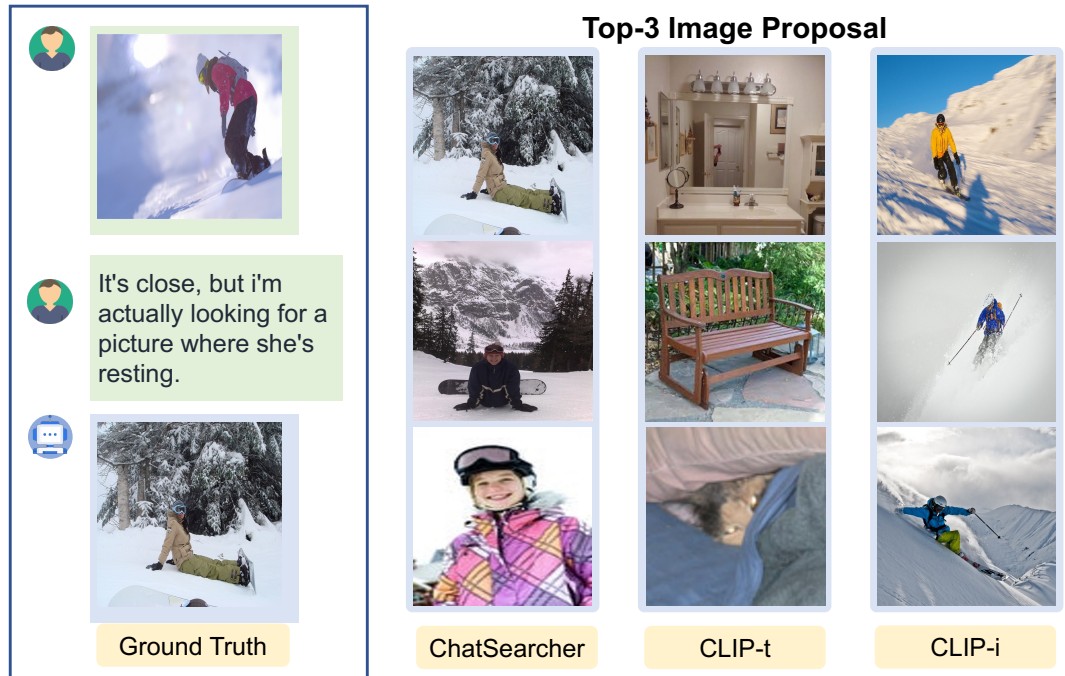

Figure 18: Visualize comparation between ChatSearcher and other methods.

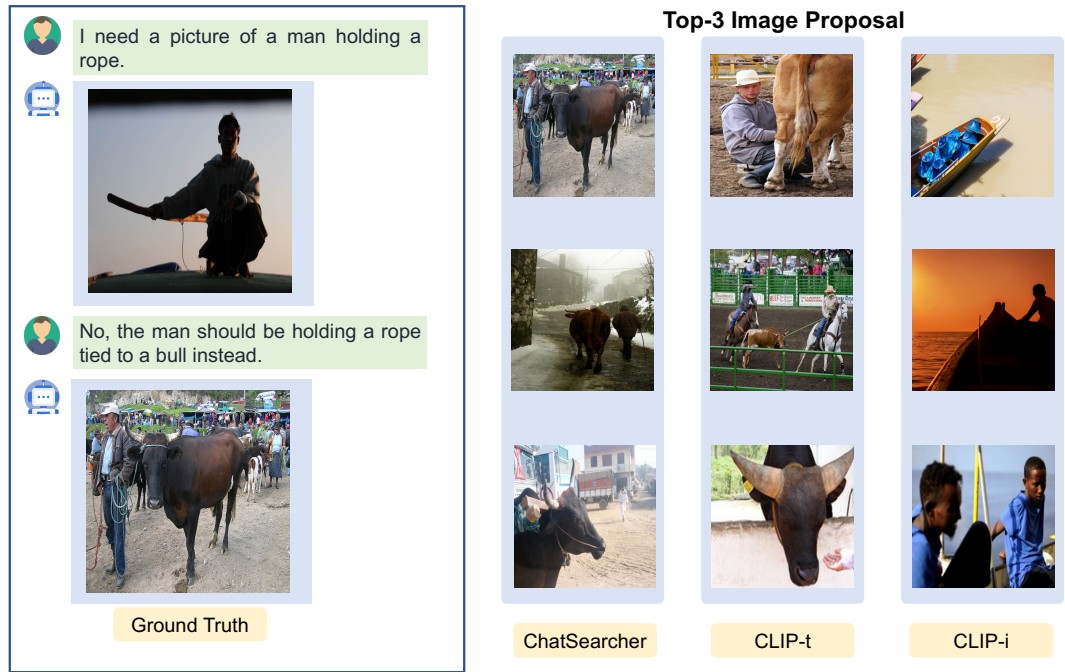

Figure 19: Visualize comparation between ChatSearcher and other methods.

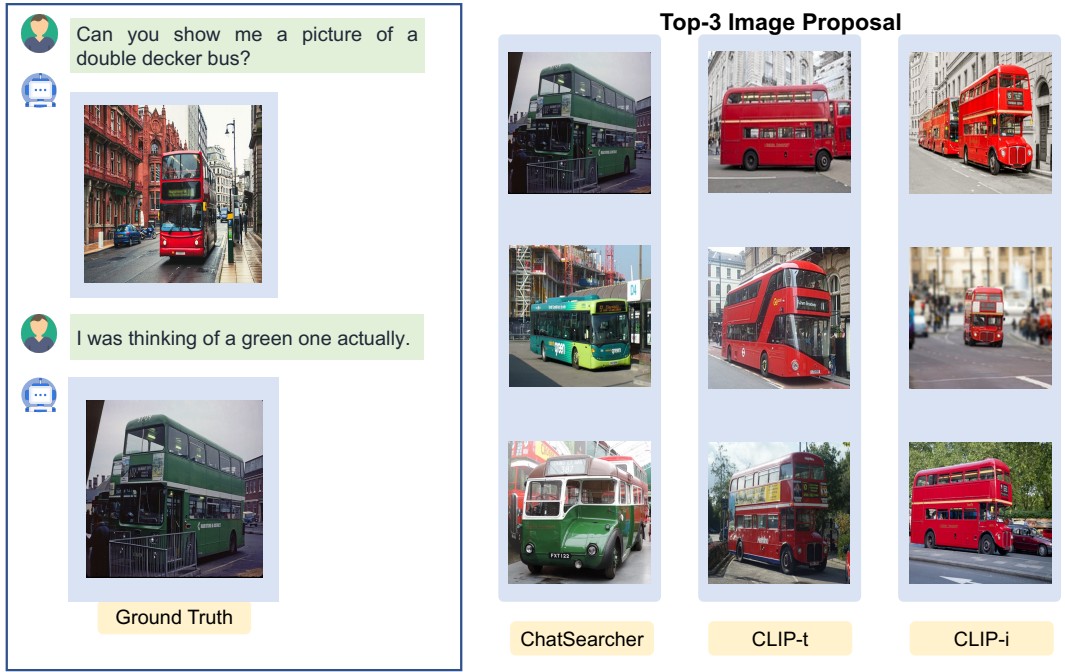

Figure 20: Visualize comparation between ChatSearcher and other methods.

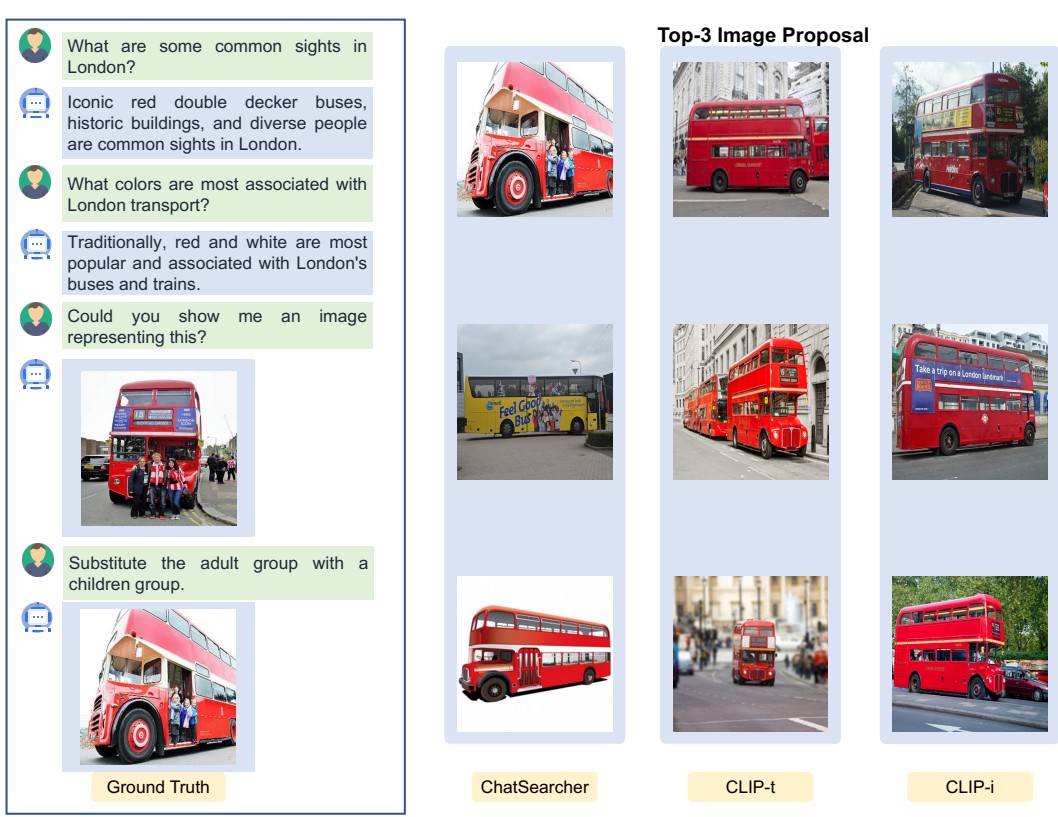

Figure 21: Visualize comparation between ChatSearcher and other methods.

