# OpenReview forum: "ChatSearch: a Dataset and a Generative Retrieval Model for General Conversational Image Retrieval"
_ICLR.cc/2024/Conference — ICLR 2024 Conference Withdrawn Submission_

### Official Review · Reviewer_nix8 · 2023-10-24

**Soundness:** 3 good
**Presentation:** 1 poor
**Contribution:** 2 fair
**Rating:** 5
**Confidence:** 3

**Summary:**

The paper presents an automatically generated dataset for conversational image retrieval, in which a user may have multi-round interaction with a retriever in a natural language. The samples in the dataset is generated using existing models. The paper also presents a model for addressing the task. Some experimental results are presented on the dataset, which draws some insights on the data required for training.

**Strengths:**

(1) I think the task can be new and interesting. If I understand the task correctly, the model is required to take the conversation history into account for better performance.

**Weaknesses:**

(1) The paper does not provide sufficient details about the task definition. The paper will be better if it provides the input and output for each task (tChatSearch, iChatSearch, and mChatSearch).

(2) Related to (1), these tasks seem to be interactive, in which a user can make different reactions to the same retrieval result (candidate images, etc.). However, as far as I understand, the dataset does not provide this type of interaction, and it only offers a single sequence of dialog without branches. This point seems important for interactive image retrieval but is not mentioned in the paper. I think this is not evaluated.

(3) I guess this is due to the page limitation, but the details on the dataset construction and the model is not fully provided to understand what is actually done. For example, the paper says some constraints are added to GPT-4 to imply some hints for image retrieval, but I cannot see how.

(4) The insufficient details on the tasks and the methods make it hard for me to evaluate the paper.

**Questions:**

I would like clarification on (1)-(3) in the weakness section. I'm happy to increase my scores once some details are provided or I have made some misunderstandings.

UPDATE: I appreciate the authors' efforts to answer my questions. I increased my score.

---

> ### Author Response · Authors · 2023-11-16
>
> Thank you for your efforts in reviewing! We have updated a new version of full paper considering all reviewers' advice. Here we provide some explanations for the weaknesses and questions raised before.
>
> **Weakness1**: The paper does not provide sufficient details about the task definition. The paper will be better if it provides the input and output for each task (tChatSearch, iChatSearch, and mChatSearch).
>
> **Response1**:
>
> In newly updated version of full paper, we add task input and output inside Table 1.
> - tChatSearch is a task that takes **multi-round textual dialogues between user and assistant** as input and searches for images that match user intents within the textual dialogue, asking the model to find the user intent in a **multi-round textual dialogue**.
> - iChatSearch is a task that takes **a user-given image and a textual user instruction** as input and searches for images that satisfy the user's intent regarding the input images, asking the model to comprehend **image and user instruction** concurrently.
> - mChatSearch is a task that takes a **multimodal dialogue containing both visual and textual contents** as input and searches for images based on the multimodal conversational context, asking the model to comprehend a **multi-round multimodal dialogue**.
>
> **Weakness2**: User can make different reactions to the same retrieval result (candidate images, etc.). However, as far as I understand, the dataset does not provide this type of interaction, and it only offers a single sequence of dialog without branches.
>
> **Response2**:
>
> In previous dialogue-based tasks, such as VisDial[1], PhotoChat[2], LLaVA-150k[3], a **single but multi-round dialogue sequence** has been shown effective to evaluate the interaction between user and assistant. Our ChatSearch also adopted this format to evaluate the interaction between user and assistant.
>
> In Appendix D.1, we give two examples to show that ChatSearcher can deal with these interaction branches, even training with dialogue sequence instruction data. These interaction branches including **choosing different images** from model's response and **giving different user instructions** to the model in the same round.
>
> We admit that more branches can bring a more comprehensive evaluation for the conversational image retrieval assistant. We plan to create more dialogue branches both on **images and textual instructions** for ChatSearch test split in our future work. Thanks for your advice!
>
> **Weakness3**:  The details on the dataset construction and the model is not fully provided to understand what is actually done. For example, the paper says some constraints are added to GPT-4 to imply some hints for image retrieval, but I cannot see how.
>
> **Response3**:
>
> In newly updated version of full paper, we add more details to make it easier to follow.
> - Modify the dataset construction illustration in Sec 3.1 with clear mathematical notations.
> - Modify the dataset construction pipeline figure Fig 2, add more illustrative details in figure and caption.
> - Add task input and output in Table 1.
> - Add discussion about unanswerable situation and give two system-level methods to solve it in Appendix A.1.
> - Add **system prompts for GPT-4 text generation** in Appendix B.
> - Add conversational instruction tuning details in Appendix C.1.
> - Add interaction branch samples in Appendix D.1
>
> [1] Das, Abhishek, et al. "Visual dialog."
>
> [2] Zang, Xiaoxue, et al. "Photochat: A human-human dialogue dataset with photo sharing behavior for joint image-text modeling."
>
> [3] Liu, Haotian, et al. "Visual instruction tuning."

---

### Official Review · Reviewer_Lx3N · 2023-10-25

**Soundness:** 3 good
**Presentation:** 3 good
**Contribution:** 2 fair
**Rating:** 5
**Confidence:** 3

**Summary:**

This paper proposes a dataset referred to as ChatSearch, which is dialogue data in terms of image, such that the dialogue agents should retrieval corresponding image to respond the question in dialogue.
For the baseline, a system referred to as ChatSearcher is also proposed to reason multi modality between image and text.

**Strengths:**

[+] This paper addresses a situation when the dialogue is composed of an image, thus this paper proposes a dataset for image retrieval based on dialogue.

[+] New task about image retrieval in terms of dialogue

**Weaknesses:**

[-] The datasets from the works below seem to assume more general situations and to have more space for multimodal reasoning. What is the difference and contributions compared to this work?

1. PhotoChat: A Human-Human Dialogue Dataset with Photo Sharing Behavior for Joint Image-Text Modeling, ACL'21

2. TikTalk: A Video-Based Dialogue Dataset for Multi-Modal Chitchat in Real World ACM Mutimedia'23

3. DialogCC: Large-Scale Multi-Modal Dialogue Dataset Arxiv'23

[-] How can we handle it if there are no images that we want? I think our questions should be more diverse than the scope of the proposed dataset covers

[-] There are many systems for text-to-image retrieval. Does this dataset address the issues that previous text-to-image retrieval systems can not perform? I mean, we can also handle the problem of image retrieval from dialogue without involving the proposed datasets by integrating text-based dialogue systems and image retrieval systems based on many image corpus. It should be more convincing if there is evidential reason that the author has to collect dataset.

**Questions:**

See above. I am almost borderline, but there is no corresponding score for it. Therefore I want to decide my evaluations after rebuttal.

---

> ### Author Response · Authors · 2023-11-16
>
> Thank you for your efforts in reviewing! We have updated a new version of full paper considering all reviewers' advice. Here we provide some explanations for the weaknesses and questions raised before.
>
> **Weakness1**: What is the difference and contributions compared to this work considering about PhotoChat, TikTalk aand DialogCC?
>
> **Response1**:
>
>  Our proposed ChatSearch dataset is different from PhotoChat, TikTalk and DialogCC. ChatSearch dataset is proposed for **conversational image retrieval tasks**, which is different from common multimodal dialogues. We mainly require the model to be capable of **visual comprehension, multi-round conversation reasoning and image retrieval** to deal with the situation in ChatSearch.
> - PhotoChat requires the model to retrieve images based on **plain textual dialogue**, while ChatSearch additionally requires the model to **comprehend visual content** and **modify previous retrieval results based on user feedback**.
> - TikTalk is a dataset that requires the model to chat with the user according to the given video, focusing more on **video comprehension** than **retrieval**.
> - DialogCC is a daily dialogue dataset, emphasizing the provision of a relevant image for each utterance in a dialogue. DialogCC is not proposed to evaluate the search ability of a retrieval model. ChatSearch dataset places more emphasis on the **search ability** of a chat assistant, requiring the model to **comprehend the entire historical conversation** and **understand the user's intent** in the current turn, rather than simply considering the **nearest utterance** to do visual matching.
>
> We have also added some relevant methods to our related works section. Thank you for your advice!
>
> **Weakness2**: How can we handle it if there are no images that we want?
>
> **Response2**:
>
>  We have added a discussion paragraph in Appendix A.1 to talk about this unanswerable situation.
> We admit that it's necessary for a retrieval system to deal with this unanswerable situation, which can enhance user-assistant interaction and improve system robustness. We provide two alternative system-level methods to in Fig 7 and Fig 8.
> - The first method is to **let the model judge** if the query is answerable. We set a similarity threshold p and subsequently apply this threshold to winnow the retrieved image candidates based on their similarities before presenting them to the user. Only those candidates exceeding the prescribed threshold are retained. If none of image candidates is usable, the model will return a **UNFOUND response** to the user. Additionally, the model will return a predefined response indicating the absence of a suitable answer for the user's query. The model will offer several image candidates with higher similarity to serve as reference points for the user as well.
> - The second method is to **ask the user to judge** whether the returned images are usable. To be precise, we add an option labeled as **UNSATISFIED**, which users can choose when none of the returned images prove satisfactory to them as a feedback during the interaction module.
>
> **Weakness3**:  There are many systems for text-to-image retrieval. Does this dataset address the issues that previous text-to-image retrieval systems can not perform? I mean, we can also handle the problem of image retrieval from dialogue without involving the proposed datasets by integrating text-based dialogue systems and image retrieval systems based on many image corpus. It should be more convincing if there is evidential reason that the author has to collect dataset.
>
> **Response3**:
>
> - The combination of a strong textual dialogue system, such as Large Language Model(LLM), together with a image retrieval system, can somehow solve part of dialogue-based image retrieval problem.  For example, we can input the whole textual dialogue into the LLM and use the search query generated by LLM to retrieve an image with a image retrieval system. Nevertheless, a strong textual dialogue system can only reason for textual contents, **lacking visual reasoning ability**. Furthermore, the above simple combination system **can not support multi-round interaction with users** since its hard to feed the retrieved result into textual dialogue system.
> - Besides, directly utilizing a multimodal LLM for image retrieval also performs bad since it can not follow human instructions about searching for an image. Thus, it's necessary to use **instruction data** such as ChatSearch to fine-tune the model, which has been widely-used on other domain[1][2][3].
> - Finally, ChatSearch is also a suitable **evaluation benchmark** for conversational image retrieval in a common domain, which can be used to assess the ability of a chat-and-search assistant.
>
> [1] Chung, Hyung Won, et al. "Scaling instruction-finetuned language models."
>
> [2] Ouyang, Long, et al. "Training language models to follow instructions with human feedback."
>
> [2] Liu, Haotian, et al. "Visual instruction tuning."

---

### Official Review · Reviewer_fGPE · 2023-10-29

**Soundness:** 3 good
**Presentation:** 4 excellent
**Contribution:** 3 good
**Rating:** 5
**Confidence:** 4

**Summary:**

This paper studies the problem of conversational image search, where the goal is to interactively retrieve images according to the human's dialogue history. To resolve this problem, the authors propose a pipeline to automatically construct a dataset to include multimodal conversational context and target retrieval candidates. Then they trained a generative retrieval model called ChatSearcher to accepted interleaved image-text context and perform search.

**Strengths:**

- The paper is clearly written with well-defined motivation and real-world downstream use case
- The dataset construction pipeline is novel and completes the main story of the paper. The quality seems okay (since the author claimed that they have human validation)
- The system built, though far from solving the task, is approaching towards the correct direction

**Weaknesses:**

- It is kind of disappointing to base the validation and test set on a ancient dataset (MSCOCO), which has been mainly used for tuning model for almost a decade. Imaging that the COCO dataset would never cover any interesting visual entity after 2015. IMO, it would be better to consider base your test set on image collections with more diverse domains, such as the visual news dataset or CC3M dataset.

- The description on conversational instruction tuning is not sufficient for reader to understand how it actually work. Particularly, that instructPix2Pix is an image editing dataset. It would be nice to show a figure that explains how data from each domain look like and how they are processed to train model.

- It also seems to me that there is a high chance where a user request can not be fulfilled within the candidate image sets. For example, when user are asking for a iPhone with foldable screen but there is no such a thing. What would the evaluation to handle such a case where it is unanswerable?  I couldn't find any discussion on it.

**Questions:**

- Why do authors emphasize your model is a generative retriever? In text retrieval (particularly entity retrieval), generative retriever usually refers to the models that generates the exact content of the document (via constrained decoding). Whereas in this case, the model is not really generating the image (but an embedding that approximates the image neighborhood). Giving the model terminology of generative retriever give the reader an impression that you are generating the image in pixel space, so that we can do exact visual matching.

- Do we have any experiments ablating the effects of each component in the instruction-tuning datasets? It seems that Table 5 is such an ablation but where is instructpix2pix?

- What is the retrieval image candidate set in each eval setting, are we only consider re-trieving against 5k (or 1k) images?

---

> ### Author Response · Authors · 2023-11-16
>
> Thank you for your efforts in reviewing! We have updated a new version of full paper considering all reviewers' advice. Here we provide some explanations for the weaknesses and questions raised before.
>
> **Weakness1**: It is kind of disappointing to base the validation and test set on a ancient dataset (MSCOCO), which has been mainly used for tuning model for almost a decade. It would be better to consider base your test set on image collections with more diverse domains, such as the visual news dataset or CC3M dataset.
>
> **Response1**:
>
> Considering that COCO is a widely-used common object dataset in the vision-language domain, we have chosen to use COCO as our image corpus in this paper. Indeed, a more diverse image collections can be adopted to generate more general data samples. Since the **data construction pipeline** proposed in our paper is easily applied to other image-caption datasets, we will explore the data construction based on a **more extensive image collection** such as CC3M in our future work. Thanks for your advice!
>
>
>
> **Weakness2**: The description on conversational instruction tuning is not sufficient for reader to understand how it actually work. Particularly, that instructPix2Pix is an image editing dataset. It would be nice to show a figure that explains how data from each domain look like and how they are processed to train model.
>
> **Response2**:
>
> We add an illustration about conversational instruction tuning in Appendix C.1. Meanwhile, we add Fig 8 to show the pre-process method of Instructpix2pix and add Fig 9 to show samples from different instruction tuning dataset.
>
> For Instructpix2pix, we simply modify the editing instruction into a more human-like textual instruction with some **pre-defined template** as shown in Fig. 9.
>
> We firstly transform all instruction data sample into **user-assistant like conversation**. Then we employ a question-answer template like “*USER :⟨question⟩ ASSISTANT :⟨answer⟩*” to convert all conversation data into unified format and use a *[image]* placeholder to represent the image content in the **multimodal sequence**. The multimodal sequence is sent into model for instruction tuning. We compute the text and image loss using **generative training objective** proposed in Sec. 4.2 on **assistant’s answers** in each round of the instruction dialogues.
>
> **Weakness3**: What would the evaluation to handle such a case where it is unanswerable? I couldn't find any discussion on it.
>
> **Response3**:
>
> We have added a discussion paragraph in Appendix A.1 to talk about this unanswerable situation.
>
> We believe that it's more fair to assert the **relative relevance** of one image over another than to definitively determine if the query is answerable by an image. Most previous academic image retrieval tasks have been proposed to evaluate this **ranking ability** of the model, asking the retrieval model to rank all candidate images according to their relevance to a given search query. As a academic evaluation dataset, ChatSearch is also proposed to evaluate the ranking ability of the model with a more complex search query hidden in a multi-round multimodal dialogue. Consequently, we still assign ground-truth image for each conversational context to ensure there is no unanswerable situation in evaluation.
>
> We also admit that it's necessary for a retrieval **system** to deal with this **unanswerable situation**, which can enhance user-assistant interaction and improve system robustness. We provide two alternative **system-level methods** in Fig 7 and Fig 8.
>
> - The first method is to **let the model judge** if the query is answerable. We set a **similarity threshold p** and subsequently apply this threshold to winnow the retrieved image candidates based on their similarities before presenting them to the user. Only those candidates exceeding the prescribed threshold are retained. If none of image candidates is usable, the model will return a **UNFOUND response** to the user. Additionally, the model will return a predefined response indicating the absence of a suitable answer for the user's query. The model will offer several image candidates with higher similarity to serve as reference points for the user as well.
> - The second method is to **ask the user to judge** whether the returned images are usable. To be precise, we add an option labeled as **UNSATISFIED**, which users can choose when none of the returned images prove satisfactory to them as a feedback during the interaction module.

---

> > ### Author Response · Authors · 2023-11-16
> > **Answer to the Questions**
> >
> > **Question1**: Why do authors emphasize your model is a generative retriever? Giving the model terminology of generative retriever give the reader an impression that you are generating the image in pixel space, so that we can do exact visual matching.
> >
> > **Answer1**:
> >
> > We claim ChatSearcher as a **generative retrieval model** for following reasons:
> > - ChatSearcher can both **generate** text response and **retrieve** images in a **generative model** based on LLM.  Different from the previous method, it unifies **visual comprehension** and **image retrieval** in a single **generative framework**.
> > - ChatSeacher is trained with **unified generative training objective** for the multimodal sequence, treating both word prediction and image retrieval as generative progresses. In word prediction, we optimize for the probability of ground-truth word prediction within the word vocabulary. For image retrieval, we maximize the probability of image feature matching within a dynamically updated image feature queue, which can be viewed as a visual vocabulary.
> > - During the inference procedure, ChatSearcher adopts a **generative inference strategy** for both text generation and image retrieval, taking all previous tokens as input and **generate the next token in an autoregressive way**.
> > - In the generating inference, ChatSearcher generates the image in **embedding space** and does visual matching with the image corpus with embedding similarity for image retrieval.
> >
> > **Question2**: Do we have any experiments ablating the effects of each component in the instruction-tuning datasets? It seems that Table 5 is such an ablation but where is instructpix2pix?
> >
> > **Answer2**:
> >
> > In Table 5, AIGC (AI-generated content) corresponds to instructpix2pix data, as indicated in Section 5.4. We have also included explanations in the caption of Table 5 in the newly updated version.
> >
> > **Question3**: What is the retrieval image candidate set in each eval setting, are we only consider re-trieving against 5k (or 1k) images?
> >
> > **Answer3**:
> >
> > The retrieval image candidate set is all ground-truth target images in each task, which is similar to image retrieval task on MSCOCO and Flickr30K. The specific number is 5k, 10k, 10k for tChatSearch, iChatSearch and mChatSearch, respectively.

---

### Official Review · Reviewer_DUdh · 2023-11-03

**Soundness:** 3 good
**Presentation:** 2 fair
**Contribution:** 3 good
**Rating:** 6
**Confidence:** 3

**Summary:**

In this paper the authors propose a new dataset ChatSearch which included multimodal conversations. The retrieval is performed by inferring details from multiple rounds of conversations. In addition, the authors also introduce a generative retrieval model ChatSearcher trained end-to-end and produces interleaved image-text inputs or outputs. Experimental results show that ChatSearcher shows superior performance on ChatSearch and comparable results on zero-shot image retrieval.

**Strengths:**

**Originality:** The authors introduce a dataset for conversational image retrieval which deals with multimodal form of dialogue. Most of the previous works focus on having image as static and textual dialogue. This limits the users ability to chat using images. The propose dataset overcomes the disadvantage. The proposed pipeline using LLMs for the dataset creation is also novel and requires less human effort.

**Quality:** In addition to the dataset, the authors also propose a strong baseline model which learns from the conversational image dataset and performs better than CLIP on standard image retrieval datasets. The ablation studies are sound and well structured.

**Weaknesses:**

**Clarity:**
-  I find the section 3.1 very difficult to follow. The words like "target image", "reference text" etc. appear multiple times and are confusing. The authors can introduce certain mathematical notations and provide sufficient examples for a better flow of the pipeline description.

- Figure-2 contains a lot of sub-figures and details. The entire pipeline is not clearly understood from the figure and caption doesn't provide sufficient context. The figure can be improved by breaking down into individual sub-figures and the text can be made clearer.

**Questions:**

1. In the text dialogue construction, how does the authors ensure that GPT-4 doesn't generate unrelated image content or repetitive text dialogue.

---

> ### Author Response · Authors · 2023-11-16
>
> Thank you for your efforts in reviewing! We have updated a new version of full paper considering all reviewers' advice. Here we provide some explanations for the weaknesses and questions raised before.
>
> **Weakness1**: The section 3.1 very difficult to follow. The authors can introduce certain mathematical notations and provide sufficient examples for a better flow of the pipeline description.
>
> **Response1**:
>
> In newly updated version of full paper, we use different mathematical notations to represent the **different image and caption** mentioned in Sec 3.1 to prevent misunderstanding. Additionally, we clarify that "reference image" and "reference text" are used to distinguish which modality is used to generate the multimodal dialogue. "Reference image" means the dialogue is generated from **a given image**, while "reference text" means the dialogue is generated from **a given caption**. And "target image" indicates the **final image** in a generated conversation, which is used to be **ground-truth image** for the conversational image retrieval task.
>
> **Weakness2**: Figure-2 contains a lot of sub-figures and details. The entire pipeline is not clearly understood from the figure and caption doesn't provide sufficient context. The figure can be improved by breaking down into individual sub-figures and the text can be made clearer.
>
> **Response2**:
>
> In newly updated version of full paper, we replace the pipeline figure with a more detailed figure Fig 2 about **dialogue construction with reference text and image** and a figure Fig 3 explaining how to perform **context merging**. We focus on explaining the process of dialogue construction with reference text and image, which represents two different data construction sources including image captions and raw images. To make it easier to follow, we use **numerical indices** to represent the execution steps.
>
> **Question1**: In the text dialogue construction, how does the authors ensure that GPT-4 doesn't generate unrelated image content or repetitive text dialogue.
>
> **Answer1**:
>
> We adopt two steps to ensure the quality of dialogue content. The first step is to **add restrictions to GPT-4 generation prompts**, asking GPT-4 to consider the round number, sentence length, semantic relevance and grammatical coherence in the generating process.These prompts can be seen in Appendix B. The second step is to add **manually review** on generated text dialogue and modify unqualified dialogue by human experts, which is mentioned in Sec 3.2.

---

### Author Response · Authors · 2023-11-16

Thanks for all reviewers' efforts in reviewing! We thank all the reviewers for their positive feedback that considers our proposed dataset and task to be meaningful and interesting (Reviewer DUdh, fGPE, Lx3N, nix8) , our data construction pipeline to be novel (Reviewer DUdh, fGPE), and our model to be effective (Reviewer DUdh, fGPE).

We have carefully considered the comments and we're glad to receive these suggestions for improving our paper.  Here, we have updated a new version of full paper considering all reviewers' advice. The main modifications include:
- Modify the dataset construction illustration in Sec 3.1 with clear mathematical notations.
- Modify the dataset construction pipeline figure Fig 2, add more illustrative details in figure and caption.
- Add task input and output in Table 1.
- Add discussion about unanswerable situation and give two system-level methods to solve it in Appendix A.1.
- Add system prompts for GPT-4 text generation in Appendix B.
- Add conversational instruction tuning details in Appendix C.1.
- Add interaction branch samples in Appendix D.1

We also highlight the main modifications in paper with blue color.

---

### Author Response · Authors · 2023-11-22
**Request for Reviewers' Reconsideration and Further Feedback**

Thank you to all the reviewers for your suggestions. We have made improvements to our paper based on your advice. As we approach the discussion phase deadline, could the reviewers please reconsider the score of our paper, or provide further suggestions? We are willing to continue addressing any questions you may have.